# A Paradigm Shift in Primary Liver Cancer Therapy Utilizing Genomics, Molecular Biomarkers, and Artificial Intelligence

**DOI:** 10.3390/cancers15102791

**Published:** 2023-05-17

**Authors:** James Moroney, Juan Trivella, Ben George, Sarah B. White

**Affiliations:** 1Division of Vascular and Interventional Radiology, Department of Radiology, Medical College of Wisconsin, Milwaukee, WI 53226, USA; 2Division of Gastroenterology and Hepatology, Department of Medicine, Medical College of Wisconsin, Milwaukee, WI 53226, USA; 3Division of Hematology and Oncology, Department of Medicine, Medical College of Wisconsin, Milwaukee, WI 53226, USA

**Keywords:** hepatocellular carcinoma, cholangiocarcinoma, liver-directed therapies, molecular biomarkers, genomics, artificial intelligence, immunotherapy, tumor hypoxia

## Abstract

**Simple Summary:**

Primary liver tumors impose significant patient morbidity and mortality with overall poor prognosis. To date, conventional therapies have provided only modest survival benefit to patients. Developments in genomics, molecular biomarkers, and artificial intelligence are introducing novel patient-centered approaches to treat primary liver tumors to improve patient survival. Recent FDA-approved immune checkpoint inhibitors Atezolizumab–Bevacizumab and Durvalumab–Tremelimumab have demonstrated improved survival outcomes and in many cases disease downstaging to curative resection. Clinical trials investigating combined immunotherapy and locoregional therapy in advanced liver disease are ongoing with promising preliminary results. Future directions in liver cancer management will likely incorporate treatment algorithms based on individualized patient molecular biomarkers.

**Abstract:**

Primary liver cancer is the sixth most common cancer worldwide and the third leading cause of cancer-related death. Conventional therapies offer limited survival benefit despite improvements in locoregional liver-directed therapies, which highlights the underlying complexity of liver cancers. This review explores the latest research in primary liver cancer therapies, focusing on developments in genomics, molecular biomarkers, and artificial intelligence. Attention is also given to ongoing research and future directions of immunotherapy and locoregional therapies of primary liver cancers.

## 1. Introduction

Primary liver cancer is the sixth most common cancer worldwide and the third leading cause of cancer-related death [1]. Hepatocellular carcinoma (HCC) and cholangiocarcinoma (CCA) comprise almost all liver cancers (75–85% and 10–15%, respectively). Early detection of liver cancers is limited, contributing to an overall poor prognosis of both HCC and CCA. Incidence and mortality are two to three times higher in men globally [1]. The highest incidence is found in southeastern Asia, where it accounts for the leading cause of death in men and women. In 2020, there were 905,677 new cases (9.5 per 100,000) and 830,180 deaths (8.7 per 100,000) globally [1]. In the United States in 2020, the overall 5-year survival for liver cancer was 29.4%. Consequently, liver cancer generates significant individual morbidity and mortality and contributes to rising global healthcare expenditure. This review explores the latest research in primary liver cancer therapies, focusing on developments in genomics, molecular biomarkers, and artificial intelligence. Attention is also given to ongoing research and future directions of immunotherapy and locoregional therapies of primary liver cancers.

### 1.1. Hepatocellular Carcinoma

The main risk factors for the development of HCC include cirrhosis, chronic hepatitis B virus (HBV), hepatitis C virus (HCV), alcohol consumption, smoking, obesity, type 2 diabetes, and consumption of aflatoxin-contaminated foods [1]. Staging and prognosis is commonly based on the Barcelona Clinical Liver Cancer (BCLC) [2]. Despite screening efforts in patients with cirrhosis, most HCC is diagnosed at advanced-stage unresectable disease. Conventional treatment options in advanced HCC include systemic and locoregional therapies, which have provided only modest survival benefit [3]. A paradigm shift in HCC management is currently underway involving target immunotherapies, which reflects advancements in comprehensive genomic profiling (CGP) and artificial intelligence (AI).

### 1.2. Cholangiocarcinoma

Although CCA is rare compared to HCC, data show that the incidence and mortality of CCA is increasing worldwide [4]. CCA is often asymptomatic in early stages, and consequently, most patients are diagnosed with advanced unresectable disease burden. The 5-year survival of unresectable CCA is 5% [5]. Risk factors include HBV and HCV infections, hepatolithiasis, nitrosamine compounds, history of primary sclerosing cholangitis and inflammatory bowel disease, and geographic risk factors such as liver fluke infections in southeastern Asia [1].

CCA tumor location impacts diagnosis and dictates treatment options. Extrahepatic CCA (eCCA) accounts for 90% of total CCA and is classified into perihilar (50%) and distal (40%) subtypes. Intrahepatic CCA (ICC) accounts for only 10% of total CCA yet is attributed to higher morbidity and mortality relative to eCCA [6]. The Liver Cancer Study Group of Japan (LCSTJ) categorizes ICC into three patterns: mass forming, periductal infiltrating and intraductal [7]. ICC remains clinically asymptomatic until advanced disease, as intrahepatic lesions are less likely to cause obstructive jaundice compared to eCCA [8]. Surgical resection of early-stage ICC is potentially curative, yet disease recurrence is prevalent [4]. The high incidence of genetic alternations in ICC pathophysiology contributes to overall poor survival. Advances in CGP and AI have uncovered potential biomarkers for targeted immunotherapy with several phase II and III clinical trials currently underway. Current studies are also evaluating combination liver-directed therapy and targeted immunotherapy with promising early results [9,10,11,12].

## 2. Pathophysiology, Genomics and Biomarkers

CGP has contributed to significant advancements in the diagnosis, management, and prognosis of liver disease. Alterations in DNA repair pathways are major precursors to oncogenesis, including deficient mismatch repair (dMMR) and high microsatellite instability (MSI-H) genes implicated in HCC and biliary tract cancers (BTCs) [13,14]. The precise mechanisms of oncogensis, disease progression and metastasis, drug resistance, and disease recurrence are complex and multifactorial. The following overview aims to characterize the molecular basis of HCC and ICC to highlight relevant factors contributing to current therapy. It is not a complete analysis of liver tumor molecular genomics and will only focus on biomarkers pertaining to the tumor microenviroment.

### 2.1. Hepatocellular Carcinoma

Liver physiology and function includes a complex immune regulatory microenvironment controlled by hepatic stellate cells (HSCs), Kupffer cells and liver sinusoidal endothelial cells (LSECs). Chronic liver injury triggers pro-inflammatory cytokine-induced tissue remodeling, increased collagen deposition, and expanded extracellular matrix, which cumulatively lead to progressive liver fibrosis and eventual cirrhosis [15]. Chronic liver injury induces HSCs and LSECs to express high levels of transforming growth factor-ß (TGF-ß), which is the most potent stimulator of fibrogenesis [15]. Similarly hepatic injury triggers HSC trans-differentiation into myofibroblast-like cells, which further promotes fibrogenesis. These myofibroblast-like cells also release α-smooth muscle actin (α-SMA) that suppresses normal T-cell immune response, contributing to hepatic immunosuppression and pro-oncogenesis [16]. Elevated alpha-SMA is a negative prognostic biomarker in HCC [17]. Through the continued release of pro-inflammatory cytokines and upregulation of growth factors, including TGF-ß, vascular endothelial growth factor (VEGF), and platelet-derived growth factor (PDGF), myofibroblasts transform into carcinoma-associated fibroblasts (CAFs) [18,19]. The translation of CAF into HCC is mediated through hepatocyte proliferation and migration, neo-angiogenesis, and inhibition of cellular apoptosis [20,21,22,23], including the upregulation of IL-6/progranulin/mTOR signaling cascade, increased expression of co-inhibitory molecules programmed cell death-1 (PD-1), programmed cell death ligand-1 (PD-L1) and cytotoxic T-lymphocyte associated antigen-4 (CTLA-4) [24,25]. This altered HCC tumor microenvironment (TME) subsequently downregulates regulatory immune checkpoints promoting immunosuppression and HCC evasion from normal tumor surveillance mechanisms [25,26,27,28,29,30]. The Golgi membrane protein-1 (GOLM1) also promotes HCC TME immune escape via upregulation of the EGFR/PD-L1 pathway and is a biomarker of HCC progression and metastasis [23,31]. Additional biomarkers associated with disease progression and poor prognosis include decreased levels of tumor necrosis factors-α (TNF-α), elevated fibrosis-4 (FIB-4), low CD8+ T-cell infiltration, and elevated infiltration of regulatory T cells (Tregs) [22,28,32].

Alpha-fetoprotein (AFP) is a 70 kD glycoprotein normally produced in the fetal liver and yolk sac that is elevated in HCC in addition to other benign and malignant processes [33]. Serum AFP has been clinically recognized for decades as a biomarker of HCC, with numerous studies investigating the utility of serum AFP in the diagnosis, treatment response, and surveillance of HCC.

Several phase I and II studies are currently investigating therapeutic agents to mitigate the progression of liver fibrosis to prevent cirrhosis and circumnavigate the development of HCC, yet no treatments are available to date [34].

An additional feature of the HCC TME is chronic diffusion-limited hypoxia. Fibrotic architectural distortion of the liver combined with neoplastic cellular hypermetabolism results in cellular oxygen demand larger than hepatic artery oxygen supply [35]. A common finding in HCC is central tumor necrosis, which is implicated in 97% of large hepatic tumors [36]. The hypoxic TME upregulates hypoxia genes, notably hypoxia-inducible factor 1 (HIF-1), which influences gene expression in glucose metabolism, growth factor expression, cellular proliferation, angiogenesis and apoptosis [37]. Barriers to effective HCC management are rooted in the complex hypoxic TME. Bristow and Hill (2008) developed a hypoxia scoring system accounting for hypoxia gene signatures that retrospectively supported an early model for HCC risk stratification [38].

Early imaging evaluation of tissue oxygenation focused on radiolabeled biomarkers, including nitroimidazoles and nucleosides that are preferentially taken up by hypoxic cells and subsequently imaged on positron emission tomography (PET) [39,40,41,42,43]. Some studies utilized these PET tissue oxygenation biomarkers to guide radiation field and dosage in radiation therapy planning [44,45]. Newer studies have implemented functional MRI mapping to quantify tissue oxygenation. Jin et al. (2010) evaluated carbogen gas-challenge blood oxygen level-dependent (CG-BOLD) MRI in rats and found an inverse correlation between tissue oxygenation and progression of liver fibrosis (r = −0.773, *p* < 0.001) [46]. Guo et al. (2012) evaluated angiogenesis and hepatic tumor size in rats using CG-BOLD MRI, demonstrating a positive correlation between ΔR2* and tumor microvsessel density (r = 0.798, *p* = 0.01) and a negative correlation between ΔR2* and tumor size (r = −0.84, *p* < 0.001) [47]. In 2015, Zhang et al. employed the first CG-BOLD MRI in patients with HCC pre- and post-TACE demonstrating variability in T2* and R2* outcome parameters, which was attributed to HCC microenvironment complexity [48]. This preliminary study did, however, show a decrease in overall HCC oxygenation post-TACE, which the authors suggested could be used to monitor treatment response. More recently, Gordon et al. (2021) used GC-BOLD MRI in a rabbit model of HCC to evaluate tumor hypoxia in Y90 radioembolization, finding a correlation between baseline ΔR2* and tumor size post-Y90 (r = 0.798, *p* = 0.002), suggesting baseline tumor hypoxia may predict Y90 treatment response [49]. Tumor oxygenation has been proposed as a non-invasive radiologic biomarker in HCC; however, methods to standardize and quantify tumor hypoxia have yet to be clinically validated. Ongoing research is focused on radiologic imaging modalities to non-invasively evaluate tissue oxygenation.

### 2.2. Cholangiocarcinoma

The pathogenesis, disease progression, and treatment resistance of ICC is mediated by a complex and diverse network of genetic alternations, which are complicated by ICC subtype heterogeneity with different molecular mechanisms of cholangiocarcinogenesis [50]. Several oncogenic pathways have been investigated reflecting variability in progenitor cells, TME, epigenetic alterations, and carcinogen exposure [51,52,53]. All ICC subtypes arise from peribiliary gland (PBG) progenitor cells in the canals of Hering with the development of mucin-producing cholangiocytes [53,54]. Integrative genomic analysis of patients with ICC identified divergent inflammatory and proliferative classes of ICC based on gene expression profiles [55]. The authors proposed that inflammatory ICC is characterized by the overexpression of STAT3 causing an upregulation of pro-inflammatory cytokines IL-4 and IL-10, whereas proliferative ICC triggers Ras–MAPK pathway activation associated with worse prognosis and lower OS (24.3 vs. 47.2 months, *p* < 0.05). Several additional genes alternations in ICC pathogenesis have been investigated including intermediate filaments (CK-7,-17,-19,-20), markers of pancreaticobiliary and gastrointestinal origin (CA19-9, mCEA, and CA125), mucins (MUC2 and MUC5AC) and tumor suppressor protein SMAD4 [56]. Several genes involved in HCC oncogenesis have also been associated with ICC including TGF-ß/Wnt and α-fetoprotein [55,57,58].

Many solid organ tumors involve dMMR gene upregulation that imparts high tumor mutational burden (TMB-H), which has been implicated in 2–10% of advanced-stage biliary tract cancers (BTC) [13,14]. Given the rarity of ICC, most research groups ICC with general BTC isocitrate dehydrogenase 1 (IDH1) mutations have been linked to 15–30% of ICC with poor prognosis [59,60]. Other well-established oncogenic mutations in treatment-resistant BTC include tumor protein 53 (TP53, 17%), cyclin-dependent kinase inhibitor 2A (CDKN2A; 15%), fibroblast growth factor receptor 2 (FGFR2, 7.4%), and human epidermal growth factor receptor (HER), which are all associated with advanced BTC and poor overall survival [61,62,63,64]. CDKN2A-positive ICC tumors are also implicated in poor prognosis with no survival benefit following curative resection over systemic chemotherapy [62]. Similarly, other studies have demonstrated a negative predictive value and worse overall prognosis in BTC associated with IDH1, BRCA1-associated protein 1 (BAP1) and polybromo 1 (PBRM1) mutations [60,62]. Lowery et al. (2018) implemented MSK-IMPACT, a targeted CGP assay, to identify associations between genetic mutations and clinical manifestations in ICC [65]. The authors found that mutations in CDKN2A and ERBB2 conferred shorter time to progression and reduced overall survival in patients receiving systemic chemotherapy for advanced ICC. Recently, ferroptosis-related gene (FRG) upregulation has been implicated in advanced ICC [50], and a novel FRG signature model was proposed to predict ICC risk stratification and prognosis [66]. Ongoing research is underway to identify ICC prognostic biomarkers; however, no definite prognostic biomarkers in ICC have not yet been clinically validated. These data suggest that the TME is poorly described and/or of weak interest in CCA.

## 3. Artificial Intelligence

Continual advances in artificial intelligence (AI), particularly involving machine-learning (ML) and deep-learning (DL) paradigms, have been utilized to aid in the diagnosis, risk stratification, and prognosis of liver cancers, particularly focused on clinical data, histopathology, and radiology imaging. When discussing ML, this refers to the development of computer systems that can “learn” from patterns in data to extrapolate finding, without requiring instructions, but rather by using algorithms and statistical models. When discussing DL, this refers to a type of ML that is able to gather more intricate higher levels of data by utilizing processing layers. The utilization of AI models in the diagnosis and treatment of HCC and ICC will ideally contribute to tailored therapeutic approaches reflecting individual patient biomarkers and genetics to improve survival. The following overview of ML and DL paradigms is focused on current applications in liver disease to date. It is not a complete analysis or characterization of ML or DL, which is beyond the scope of this review.

### 3.1. Hepatocellular Carcinoma

AI algorithms have been designed to aid in the diagnosis of HCC. Sato et al. (2019) developed a novel ML system predictive model for HCC diagnosis using patient clinical datapoints, including alpha-fetoprotein and Des-gamma carboxyprothrombin, which demonstrated 87.34% predictive accuracy and area under the curve of 0.940 [67]. Similarly, Ksiazek et al. (2019) developed a support vector machine and genetic optimizer training ML model incorporating qualitative and quantitative criteria, including patient demographics, laboratory datapoints, and disease comorbidities, to predict the development of HCC with 88.5% yield accuracy [68].

A relatively new application of AI is focused on predictive models of HCC recurrence and survival based on several factors, including HCC genomic expression, histological features, and radiological biomarkers. Chaudhary et al. (2018) built a DL model to differentiate survival in HCC subpopulations utilizing RNA sequencing data from the Cancer Genome Atlas [69]. Their model determined that HCC subtypes with worse prognosis and lower survival are associated with TP53 mutations, elevated KRT19 and EPCAM stemness expression, BIRC5 tumor marker, and activated Wnt and Akt signal pathways.

Several studies have demonstrated the accuracy of AI histopathology models in the diagnosis of non-alcoholic hepatic steatosis (NASH) and non-alcoholic fatty liver disease (NAFLD) [70,71]. An early study by Vanderbeck et al. (2014) showed a support vector machine (SVM) algorithm to quantify hepatic steatosis with 89% accuracy [72]. More recently, Forlano et al. (2020) employed an ML quantification of liver morphology to calculate a NASH score with 80% accuracy [73]. Similarly, Gawrieh et al. (2020) developed a ML model to classify patterns of liver fibrosis with area under the curve (AUC) between 0.77 and 0.95 [74]. A landmark study by Taylor-Weiner et al. (2021) developed an ML model to quantify fibrosis and predict the progression of NASH (C-index up to 0.73) [75].

AI histopathology models have also been developed to diagnose HCC and predict survival following HCC resection. Aatresh et al. (2021) created a DL model titled LiverNet for the automatic diagnosis of HCC subtypes with 90.9% accuracy [76]. Additionally, several studies have developed DL algorithms to grade HCC by hepatic nuclei features [77,78]. Sailard et al. (2020) employed a DL model using whole-slide digitized histological slides following HCC surgical resection to predict survival [79]. This model determined that HCC tumors involving vascular spaces, macrotrabecular architectural pattern and lack of immune infiltration were most predictive of poor survival outcome (C-index up to 0.78). Another ML model using digital histopathology predicted early HCC recurrence following surgical resection with 90% accuracy [80].

Two categories of AI have been utilized in the imaging of liver disease. Radiomics uses supervised ML algorithms to quantify imaging features into relevant datapoints, and convolutional neural networks (CNNs) employ automated DL systems to extract pertinent imaging parameters [81]. Both AI tools have been devised to detect liver lesions, classify lesion severity, and predict survival outcomes.

Early work by Christ et al. (2016) used DL model CNNs to map liver morphology and identify focal liver lesions on CT [82]. Hasan et al. (2017) employed DL stacked sparse auto-encoding in ultrasound to classify liver lesions as cysts, hemangiomas, or HCC by highest probability with 97.2% accuracy [83]. DL systems have also been used in an ultrasound to categorize focal liver lesions as benign or malignant with a mean receiver operating characteristic of 0.916 [84] and accuracy, sensitivity, and specificity of 93%, 91%, and 97%, respectively [85]. Additional AI systems employing CT and MR modalities to identify and categorize focal liver lesions have yielded similar results [86,87,88,89]. Traditional computer-aided diagnostic systems relied on manual data acquisition from imaging features including texture and contour segmentation to characterize and classify tumors. These models provided similar high accuracy in tumor detection to current DL and ML models, yet they relied on supervised rather than automated data acquisition and longer processing times [82,84,86].

Microvascular invasion (MVI) is a major predictor of tumor recurrence post HCC resection, yet no clinically validated predictive parameters of MVI have been widely implemented into routine patient management. Several studies have reported promising early results, including Dong et al. (2020), who developed a radiomic AI algorithm using grayscale ultrasound to predict pre-operative HCC MVI [90]. The model quantified US radiomic signatures of gross-tumoral region (GTR), peri-tumoral region (PTR), and gross peri-tumoral region (GPTR), which combined with clinical datapoints classified pre-operative HCC patients into low-risk MVI and high-risk MVI groups with up to 81% accuracy. Similarly, Ji et al. (2019) used an ML algorithm of contrast-enhanced CT to develop three radiomic signatures to predict HCC recurrence in combination with clinical datapoints, which demonstrated promising prognostic probability for HCC recurrence (C-index of 0.733–0.801 and integrated Brier score of 0.147–0.165; *p* < 0.05) relative to prior non-radiomics models including the Early Recurrence After Surgery for Liver (ERASL) model (C-index of 0.622; *p* < 0.001) [91]. A similar multiphase-CT DL model was developed by Wang et al. (2019) yielding an area under the curve of 0.825 [92]. A recent study by He et al. (2021) used a DL model to incorporate MRI radiomics, histopathology, and clinical datapoints to predict HCC recurrence post-liver transplant with promising results (AUC 0.87) [93].

AI models have also been developed to predict response to liver-directed therapy. Abajian et al. (2018) developed an ML model based on MRI radiomics and clinical datapoints to predict treatment response to TACE with 78% accuracy and 88.5% negative predictive value [94]. The model determined that the strongest predictors of post-TACE treatment response were the presence of cirrhosis with elevated relative tumor signal intensity. Similar results were obtained evaluating treatment response following radiofrequency ablation [95]. Morshid et al. (2019) predicted post-TACE response using a DL model combining CT radiomics with BCLC score, which demonstrated higher accuracy relative to BCLC stage alone (74.2% and 62.9%, respectively) [96]. Several DL models have also successfully predicted post-TACE response using pre-treatment CT radiomics, RECIST criteria and MVI [97,98].

### 3.2. Cholangiocarcinoma

ICC is a rare disease with complex genetic alterations that affect disease progression and treatment resistance. Most studies rely on pooled data using all BTC subtypes to evaluate a treatment regimen. To date, there is no standardized staging or treatment algorithm to identify patients who will benefit from conventional systemic therapy, immunotherapy, locoregional therapy, or combination therapy [99]. Consequently, AI models are being developed to assist in identifying ICC risk factors, diagnosing, and staging of disease and predicting survival outcomes based on genetic markers [100].

Ji et al. (2022) developed a gradient boosting machine learning (GBM) model to predict the likelihood of cancer-specific survival following ICC surgical resection [100]. The performance of the GBM model was compared to the prior prognostic score and staging systems for ICC that have shown only modest prognostic accuracy [92,101]. The multifocality, extrahepatic extension, grade, nodal status, and age (MEGNA) staging system and the American Joint Committee on Cancer (AJCC) ICC staging manual both rely on data acquired following post-operative hepatectomy and are restricted by qualifiers that preclude modification in the assessment tool. The GBM model was trained using data from an international cohort of over 1000 patients with ICC. The authors reported that the GBM model outperformed both the AJCC and MEGNA models in identifying low-risk and early-stage ICC patients [100]. The GBM model also established a risk stratification tool for cancer-specific-death based on low, intermediate, and high-risk groups.

Zhou et al. (2022) similarly developed an ML model to establish a clinically relevant support tool to predict outcomes in patients with ICC [102]. The ML model was developed by harvesting data from over 4000 patients from the surveillance, epidemiology, and end results (SEER) database. The authors report that the ML model successfully predicted short-term prognosis in ICC patients following treatment based on clinical parameters, surgery, systemic therapy, and TNM staging.

The incorporation of AI applications in healthcare is rapidly evolving and will likely become a significant contributor to the diagnosis, risk stratification, and treatment of liver disease and beyond. Ideally, AI will help contribute to tailored therapeutic approaches in HCC and ICC reflecting individual patient biomarkers and genetics.

## 4. Treatment

### 4.1. Hepatocellular Carcinoma

Patients with localized and early-stage HCC may undergo curative surgical resection or locoregional therapy, but advanced disease is treated with systemic therapy. Despite the wide availability of conventional therapies for HCC, overall survival rates remain poor due to treatment resistance, high disease recurrence and metastasis [103], including an estimated 70–80% disease recurrence in early-stage HCC following curative resection or ablation [104].

Early systemic first-line treatment of advanced HCC was Sorafenib, an anti-VEGF/anti-PDGFR tyrosine kinase inhibitor, which in the SHARP trial demonstrated superior OS relative to placebo (median OS 10.7 vs. 7.9 months) [105]. The REFLECT phase III trial in Japan subsequently demonstrated that Lenvatinib increased survival compared to Sorafenib (median OS 13.6 vs. 12.3 months, HR = 0.92, 95% CI: 0.79–1.06) [106]. More recently, studies have investigated the efficacy of immune checkpoint inhibitors (ICIs). Pembrolizumab (anti-PD-1 mAb) was FDA approved for the treatment of unresectable or metastatic MSI-H/dMMR solid tumors including as second-line treatment of advanced HCC following results from the KEYNOTE-240 phase III trial that demonstrated a trend toward improved median PFS and median OS despite not reaching statistical significance [107]. Similarly, the CHECKMATE 459 randomized, multicenter, open-label phase III trial reported no significant difference in overall survival between Nivolumab and Sorafenib, yet Nivolumab has potential for treatment in patients with contraindications to Sorafenib. Furthermore, grade 3 or 4 adverse events were reported in 22% for Nivolumab compared to 49% in Sorefenib [108]. A list of completed and ongoing trials is summarized in Table 1.

The IMBRAVE150 global, open-label, phase III trial demonstrated that the combination of Atezolizumab (Anti-PD-L1 mAb) and Bevacizumab (anti-VEGF) relative to Sorafenib resulted in superior overall survival (median OS 67.2% vs. 54.6%) and improved median PFS of 6.8 months vs. 4.3 months (HR = 0.59, 95% CI: 0.47–0.76, *p* < 0.001) [107]. In 2021, the IMBRAVE150 update with an additional 12-month follow-up confirmed survival benefit with a median OS 19.2 vs. 13.4 months (HR 0.66, 95% CI 0.52–0.85, *p* < 0.001) and updated ORR of 29.8% (per RECIST 1.1 criteria) including complete response in 7.7% of patients [109]. Atezolizumab–Bevacizumab subsequently became first-line treatment for unresectable HCC. Additionally, a second drug combination was recently FDA-approved: Durvalumab/Tremelimumab (DT) immunotherapy following results from the HIMALAYA phase III trial which demonstrated significantly increased survival relative to Sorafenib with a median OS of 16.4 months vs. 13.8 months (HR 0.78, 95% CI 0.66–0.92, *p* = 0.0035), median PFS of 3.8 months (95% CI: 3.7–5.3) vs. 4.1 months (95% CI: 3.7–5.5) and ORR of 20.1% (95% CI: 16.3–24.4) vs. 5.1% (95% CI: 3.2–7.8) [110].

Liver-direct locoregional therapies (LRTs) utilize intra-arterial embolization and percutaneous ablative techniques to treat HCC for bridge to transplant or curative intent in early-stage disease, for downstaging to resection or transplantation in intermediate-stage disease, and for palliation in advanced-stage disease. The effect of LRTs on HCC is twofold: via direct damage of tumor cells and by immunomodulation of the TME. Induced tumor necrosis following LRT triggers a local and systemic immune response, which has been shown to enhance the anti-tumor response but paradoxically may also stimulate oncogenesis via pro-inflammatory triggers, overexpression of hypoxia-induced factors (HIFs), VEGF upregulation of angiogenesis, and tumor cell metastasis [103,111,112,113]. Studies have reported HCC tumor progression, disease recurrence, and overall worse outcomes in both thermal and non-thermal liver-directed LRTs [35,103,112,113,114,115,116]. Immunomodulation following LRTs has also demonstrated improved overall survival via the activation of intratumoral infiltrates including tumor-specific CD8+ T-cells [117,118,119,120,121,122,123]. Combination LRT and targeted immunotherapy studies to evaluate potential synergistic anti-tumor responses are currently underway.

Radiofrequency ablation (RFA) is the first-line ablative therapy used for early-stage HCC in tumors < 5 cm with high efficacy and low risk of complications [103,124,125], but it has been associated with tumor progression and recurrence when used as monotherapy [124], possibly reflecting incomplete ablation zones [126], resulting in elevated HIF-1 and VEGF levels that have been associated with worse prognosis following RFA [127]. However, the incorporation of volumetric ablative margin registration software following RFA significantly improved assessment of HCC ablation zone completeness with better predictive value for local tumor progression and lower tumor recurrence [128,129,130,131]. RFA-induced local and systemic immunomodulation also increases tumor-specific CD8+ T cells [117,121] and decreases pro-oncogenic factors TGF-ß, IL-10, and Tregs [123,132,133]. Increased infiltrating CD45RO+ memory T cells following RFA is a biomarker for improved clinical outcomes in solid tumors [134]. Several studies have investigated combination RFA and targeted immunotherapy in HCC with preliminary results showing improved anti-tumor T cell response, reduced risk of HCC recurrence and improved PFS [135,136,137,138,139]. Recently published interim data from the IMBRAVE050 phase III, multicenter, randomized, open-label clinical trial demonstrated adjuvant Atezolizumab–Bevacizumab therapy following early-stage HCC curative resection or ablation (RFA or MWA) significantly improved recurrence-free survival relative to active surveillance [104].

TACE is the standard treatment for a subset of patients with intermediate-stage HCC reflecting the recent 2022 Barcelona Clinic Liver Cancer (BCLC) update indicating well-defined multinodular HCC with patent portal vein and preserved liver function [2,103] (see Figure 1). TACE causes local tumor cell destruction and induces a systemic immunomodulating response leading to elevated CD4+/CD8+ ratio, increased NK cells, decreased Treg, and decreased CD8+ T cells [140,141,142]. Similarly to RFA, studies have shown that post-TACE elevation in HIF-1 and VEGR is associated with poor prognosis [103,143]. Patients with advanced-stage HCC have limited treatment options. TARE is used to for tumor downstaging to transplant or resection or for palliation [144]. TARE similarly activates a strong systemic immune response with elevation in infiltrating CD8+ T cells, NK cells, and TNF-α [145,146]. A meta-analysis by Zhang et al. (2015) comparing outcomes of TACE and TARE in unresectable HCC showed that TARE was associated with less HIF-1 and VEGF upregulation and improved overall survival [147]. Several current studies are evaluating the effect of IAT combined with ICI with preliminary data suggesting an augmented immunological response leading to improved survival outcomes. An open-label, Phase I study that evaluated TARE and Nivolumab reported an 82% disease control rate and that 46% of patients had decreased circulating AFP levels [148]. A phase II trial of TARE followed by Nivolumab demonstrated improved outcomes with ORR 30.6% (95% CI: 16.4–48.1). The NASIR-HCC phase II trial of Nivolumab and TARE reported ORR of 41.5% (95% CI: 26.3–57.9%) and mean OS of 20.9 months (95% CI: 17.7–24.1) with four patients downstaged to resection [149]. Additional trials investigating Nivolumab with DEB-TACE and Pembrolizumab with TACE are underway [150,151].

The image of a contrast-enhanced CT on the left shows a cirrhotic liver with arterial phase hyperenhancing segment 7 mass, corresponding to a known HCC. A non-contrast CT image on the right is post-TACE showing residual lipiodol in the right hepatic lobe.

### 4.2. Cholangiocarcinoma

Systemic therapy is the only available treatment for patients with unresectable and recurrent ICC. First-line treatment with combination gemcitabine and cisplatin (GC) was based on the benchmark ABC-02 phase III trial by Valle et al. (2010) that demonstrated a 36% reduced risk of disease progression relative to gemcitabine alone (hazard ratio = 0.65, *p* < 0.001); however, OS remained dismal at 11.7 months [152]. Similar results were obtained in the Japanese BT22 phase II study [153]. Subsequent studies provided clinical validation for CG treatment in advanced biliary tract cancers and are considered standard of care despite only modest survival benefit [154]. Furthermore, not all patients with ICC respond to CG combination therapy. Argarwal et al. (2016) found that patients with poor performance status (PS > 2), elevated CEA (>3) and advanced disease (Stage IVb) at baseline were negative prognostic predictors of poor response to GC therapy [154]. Research into neoadjuvant chemotherapy regimens for ICC is ongoing, including the SWOG-1815 phase III trial investigating triple therapy gemcitabine, cisplatin and nab-paclitaxel, with promising results from its phase II trial showing 11.8 month progression-free survival (95% CI: 6.0 to 15.6 months) and overall survival 19.5 months (95% CI: 10.0 months to non-estimable) [155]. The authors also reported that 20% of subjects achieved downgraded disease burden and underwent curative resection.

The IDH1 inhibitor ivosidenib was recently evaluated in the ClarIDHY phase III multicenter, randomized, double-blind study, which compared ivosidenib to placebo in IDH1-mutant ICC demonstrating significantly improved progression-free survival (2.7 months vs. 1.4 months; HR = 0.37, *p* < 0.001) and median overall survival (10.3 months vs. 5.1 months; HR = 0.49, *p* < 0.001) [156,157]. Several studies have demonstrated mixed results evaluating the efficacy of EGFR inhibitors, which block the effect of ERBB2 mutations in ICC, including monotherapy with erlotinib and cetuximab, in addition to combination therapy of EGRF inhibitors with gemcitabine–cisplatin [158,159,160]. FGFR inhibitors have been evaluated in phase II trials for advanced ICC [161,162,163]. The FIGHT-202 phase II trial demonstrated improved clinical benefit from pemigatinib with ORR 35.5% (95% CI: 26.5–45.4) and it is currently approved for advanced BTCs in many countries for patients with FGFR2 fusion or rearrangement [63].

Molecular biomarkers identified from ICC genetic sequencing data have been used as targets of immunotherapy in advanced ICC, including a list of completed and ongoing trials summarized in Table 2. Monotherapy with PD-1 and PD-L1 inhibitors has demonstrated only a modest benefit in patients with advanced BTCs with objective response rates (ORRs) between 3 and 7% in the largest clinical trials [164,165]. A KEYNOTE-158 Phase II trial evaluating Pembrolizumab (anti-PD-L1) in treatment-resistant advanced BTC demonstrated median PFS 2.0 months (CI 1.9–2.1) and median OS 7.4 months (CI 5.5–9.6) [164]. In the study, 64% of the 95 patients had PD-L1 tumor expression with an objective response rate (ORR) by RECIST to Pembrolizumab of 6.6% (CI 1.8–15.9) in PD-L1 positive patients compared to 2.9% (CI 0.1–15.3) in PD-L1 negative patients. Of note, none of the patient had MSI-H mutations. Few patients experienced grade 3 adverse events (13%), and no grade 4 or 5 adverse events were reported. Subsequently, Pembrolizumab was FDA approved for the treatment of unresectable solid tumors with MSI-H or dMMR based on data accumulated from five KEYNOTE clinical trials [166]. Therefore, the result of immunotherapies is disappointing and far worse than those reported with TACE and TARE when used in isolation. This has led to trials evaluating combination therapies.

Several newer studies have reported a cumulative benefit from combination therapy. Feng et al. (2020) conducted a phase II trial and reported that 1st-line CG systemic chemotherapy combined with Nivolumab immunotherapy led to a median PFS of 6.1 months, median OS of 8.5 months and a 33% 12-month OS rate [167]. Klein et al. (2020) demonstrated an ORR of 23% with Nivolumab combined with anti-CTLA-4 drug Ipilimumab [168]. Combined anti-MEK (Cobimetinib) and anti-PD-L1 (Atezolizumab) immunotherapy in a phase II trial of BTC found that a reduction in platelet-derived growth factor B (PDGF-BB) was associated with increased overall PFS yet was negatively correlated with OS [169]. Research has also focused on the combined effects of Durvalumab (anti-PD-1/PD-L1 mAb) with Tremelimumab (anti-CTLA-4 mAb) (DT) and conventional GC therapies. A phase I trial by Doki et al. (2022) compared DT to Durvalumab alone in advanced BTC demonstrating greater benefit with combination DT therapy (ORR 10.8% vs. 4.8%) [170]. An active open-label, single-center, phase II trial by Oh et al. (2022) is evaluating the efficacy of GC with DT compared to GC with Durvalumab alone in patients with advanced BTC, with initial results showing improved response rates relative to GC alone (ORR 72%, 70%, and 50%, respectively) [171]. The TOPAZ-1 double-blind, placebo-controlled, phase III trial reported that GC with Durvalumab significantly improves survival relative to GC alone with OS 24.9% (95% CI 17.9–32.5) vs. 10.4% (95% CI 4.7–18.8; *p* + 0.001), and ORR 26.7% vs. 18.7% [172], and it has become the standard first-line treatment for unresectable BTC.

Unresectable ICC treated with palliative systemic chemotherapy and radiotherapy confers high levels of toxicity. Alternatively, locoregional liver-directed therapies have demonstrated survival benefit in unresectable ICC with fewer toxicities and with the potential for downstaging to curative resection [11,173,174,175]. Numerous studies and meta-analyses have demonstrated improved survival benefit in unresectable ICC following TACE with comparable results in conventional TACE (cTACE) and drug-eluting bead TACE (DEB-TACE) [173,176,177,178]. An early retrospective study by Vogl et al. (2012) demonstrated increased median OS of 13 months and survival rates of 52%, 29%, and 10% at 1, 2, and 3 years, respectively [176]. Another early retrospective study showed improved overall survival following cTACE relative to supportive care (12.2 months vs. 3.3 months, *p* < 0.001) [179]. Meta-analyses have similarly demonstrated a significant survival benefit post-TACE relative to systemic chemotherapy with fewer adverse events and drug toxicities [177,178]. ICC tumor downstaging following TACE has also been reported in a small percentage of patients [175,180].

Other studies have evaluated the benefit of TARE in unresectable ICC, demonstrating prolonged survival [101,181,182]. Mouli et al. (2013) reported improved overall survival and disease downstaging following TARE, with 11% of patients downstaged to surgical resection with 100% survival at 2.5-year follow-up [183]. The authors also showed a stratified survival benefit dependent on disease burden, comparing multifocal and solitary tumors (14.6 vs. 5.7 months), infiltrative (6.1 vs. 15.6 months), and bilobed disease (10.9 vs. 11.7 months). A systematic review of 12 studies showed a mean OS of 17.7 months TARE with 10% of patients downstaged to surgical resection [184]. In 2021, a systematic review and meta-analysis of 31 studies by Mosconi et al. demonstrated comparable overall survival benefit between TARE and TACE (mean OS 13.5 vs. 14.2 months) with fewer toxicities following TARE [178].

Newer research has focused on the benefit of combined locoregional and systemic therapies to treat unresectable ICC, reflecting radiosensitizing chemotherapeutic amplification of TARE [185]. An early study by Rayar et al. (2015) showed promising results of increased median disease-free survival of 19.1 months following a combination of systemic chemotherapy (gemcitabine and/or platinum salts) followed by TARE, albeit results were limited by a small sample size [10]. More recently, a small phase Ib study of eight patients with unresectable ICC (*n* = 5) and hepatic metastasis from pancreatic cancer (*n* = 3) received gemcitabine followed by TARE demonstrated a median hepatic PFS of 20.7 months for ICC patients [9]. The MISPHEC phase II trial evaluated systemic gemcitabine and cisplatin chemotherapy with concomitant Y-90 radioembolization in unresectable ICC. The results showed a median PFS of 14 months and median OS of 22 months, with 22% of patients downgraded to surgical intervention [11]. These results demonstrated an OS was doubled compared to results of systemic therapy alone in the ABC-02 trial. Currently, a phase II trial is evaluating combination Durvalumab/Tremelimumab with concomitant radiation therapy with promising preliminary results [186].

## 5. Conclusions

Primary liver cancers continue to impose significant morbidity and mortality worldwide with overall poor prognosis. Conventional systemic and locoregional therapies for advanced-stage disease have provided only modest survival benefit. A paradigm shift in liver cancer management is currently underway, reflecting molecular biomarker-targeted immunotherapy based on advancements in comprehensive genomic profiling and artificial intelligence. Recent FDA-approved immune checkpoint inhibitors Atezolizumab–Bevacizumab and Durvalumab–Tremelimumab have demonstrated improved survival outcomes and in many cases disease downstaging to curative resection. Clinical trials investigating combined immunotherapy and locoregional therapy in advanced liver disease are ongoing with promising preliminary results. Future directions in liver cancer management will likely incorporate treatment algorithms based on individualized patient molecular biomarkers.

## Figures and Tables

**Figure 1 cancers-15-02791-f001:**
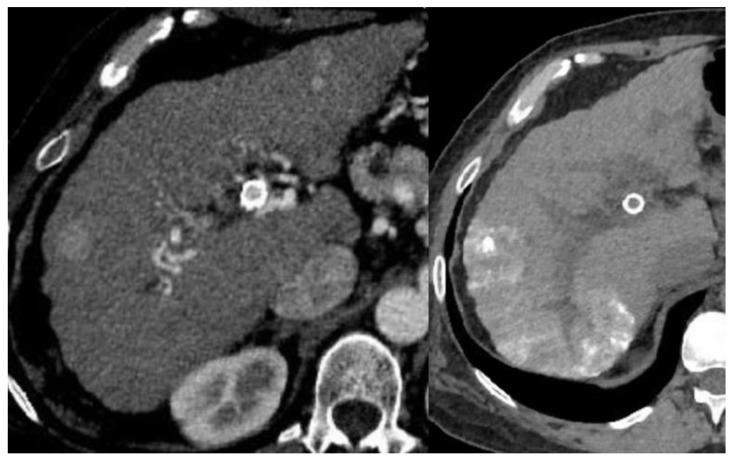
Hepatocellular carcinoma pre- and post-TACE.

**Table 1 cancers-15-02791-t001:** Clinical trials involving immunotherapy for hepatocellular carcinoma treatment.

Trial	Immunotherapy	Biomarker Target	Outcome	Sample Size
KEYNOTE240	Prembrolizumab	PD-L1	PFS, OS	413
CHECKMATE459	Nivolumab	PD-1	OS	743
IMBRAVE150	Atezolizumab plus bevacizumab	PD-L1, VEGF	OS	501
HIMALAYA	Durvalumab plus tremelimumab	PD-L1/CTLA-4	OS	1504
IMBRAVE050	Atezolizumab plus bevacizumab (following resection or ablation)	PD-L1	PFS	662
NASIR-HCC	Nivolumab (following TARE)	PD-1	ORR, TTP, OS	42
EMERALD-2	Durvalumab plus bevacizumab	PD-L1	PFS	908
	Nivolumab (following DEB-TACE)	PD-1	Safety and efficacy	20
PETAL	Prembrolizumab (following TACE)	PD-L1	Safety and efficacy	14

PFS: progression-free survival; OS: overall survival; ORR: objective response rate; TTP: time to progression.

**Table 2 cancers-15-02791-t002:** Clinical trials involving immunotherapy for biliary tract cancers.

Trial	Immunotherapy	Biomarker Target	Outcome	Sample Size
KEYNOTE158	Prembrolizumab	PD-L1	PFS, OS	104
TOPAZ-1	Gemcitabine and cisplatin plus Durvalumab	PD-L1	OS, ORR	810
	Nivolumab plus gemcitabine and cisplatin	PD-1	Safety and efficacy	32
	Nivolumab and Ipilimumab	PD-1/CTLA-4	Safety and efficacy	39
	Gemcitabine and cisplatin plus Durvalumab with or without Tremelimumab	PD-L1/CTLA-4	Safety and efficacy	128
MED14736	Durvalumab plus Tremelimumab (plus radiation therapy)	PD-L1/CTLA-4	Safety and efficacy	70

PFS: progression-free survival; OS: overall survival; ORR: objective response rate.

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
