# Peer review of "A Paradigm Shift in Primary Liver Cancer Therapy Utilizing Genomics, Molecular Biomarkers, and Artificial Intelligence"

_cancers, 2023, doi:10.3390/cancers15102791_

Round 1
Reviewer 1 Report
A Paradigm Shift in Primary Liver Cancer Therapy Utilizing Genomics, Molecular Biomarkers, and Artificial Intelligence by Moroney presents an overview of genomics, molecular markers and AI in HCC and CCA by Moroney et al submitted to Cancers.
Comments
If authors could include a summary table with a list of most prevalent gene mutations and current treatments for HCC and ICC with a reference would be good. Also please provide brief descriptions on the mechanisms of how these mutations are known to cause liver cancer progression, recurrence, drug resistance, metastatic behavior etc. Also, if authors could provide their own thoughts and perspective on how AI could help in establishing personalized medicine to treat HCC and CCA would be good.
Acceptable. Minor editing is required.
Author Response
Thank you for your review, I appreciate your time and suggestions.
A table summarizing relevant gene mutations and corresponding treatments for HCC and ICC will be added to the manuscript.
The precise mechanisms of oncogensis, disease progression and metastasis, drug resistance, and disease recurrence are complex, heterogenous and multifactorial. The overview in the manuscript is intended to characterize the molecular basis of HCC and ICC to highlight relevant factors contributing to current and ongoing therapy. It is not a complete analysis of HCC and ICC molecular genomics, which is beyond the scope of the paper.
A brief description of the effect of AI on personalized treatment has been added to the manuscript.
Reviewer 2 Report
REVIEW OF ARTICLE ENTITLED A Paradigm Shift in Primary Liver Cancer Therapy Utilizing 2 Genomics, Molecular Biomarkers, and Artificial Intelligence
Abstract: Representative of the article
Introduction: Diligent, well written and well cited. Should the authors provide information obtained from tumor biopsy compared with plasma factors in humans? Do the authors recommend liver tumor biopsy prior to treatment ?
Pathophysiology, genomics, radiomarkers: See also above.
Treatment: Should the authors comment on the value of chemoembolization as a primer to immunotherapy or combination of the two treatments in HCC ? Short mention of complications and eligibility for immunothery are advised to be included. In addition could the authors state the point where immunotherapy should stop? Recurrence, increase of biomarkers etc?
Conclusion: Well written
References: Relevant and adequate
Author Response
Thank you for your review. I appreciate your time and suggestions.
Regarding tumor biopsy. Most patients do not receive a biopsy prior to initiating treatment, especially when imaging characteristics are classic for HCC and ICC. I agree that a comparison of tissue sampling results with plasma biomarkers can can provide information about sensitivity and specificity of biomarkers. I will add a brief description about this.
A description of immunotherapy eligibility and complications will also be added.
Regarding combination chemo-embolization and immunotherapy. Since numerous clinical trials are ongoing to evaluate the efficacy of these treatments in parallel compared to sequentially, it is yet to be determined if one approach is more effective.
Reviewer 3 Report
Reviewing report for manuscript for Cancers: “A paradigm shift in primary liver cancer therapy utilizing genomics, molecular biomarkers, and artificial intelligence” form James Moroney et alii.
In this review Moroney and coworkers provide a start of the art of genomic biomarkers, artificial intelligence and therapy of primary liver cancer ie hepatocellular carcinoma (HCC) and cholangiocarcinoma (CCA). The program is, thus, particularly vast and the work intrinsically ambitious as it could be treated in length and without losing interest by many different papers.
The authors have to be quick and to make choices offering (boldly) their flank to criticisms. However, the paper is pleasant to read as it represents a synthesis of almost 200 references and enable the reader to get insights about the states of the art in different domains.
As a biologist, the first part dedicated to genomics and biomarkers is the one that could generate the largest amount of criticisms especially for HCC as choices correspond somehow to look at this particularly heterogeneous tumor through the keyhole. I will thus recommend to add a sentence stating that the authors focused on biomarkers pertaining to the tumor microenvironment (TME) only. Curiously, for CCA, markers description is classical (TP53, IDH1, CDKN2A, ERBB2, etc). A situation that implies the fact that TME is poorly described or of weak interest in CCA. A sentence to make it clear will be welcomed as well.
Regarding Artificial intelligence and therapy, tables recapitulating main findings are necessary.
Definition of abbreviations, very numerous in the paper, is necessary in a separate abbreviation list eg “a targeted CGP assay” (line 173)
Quick definitions of the nature and differences between machine learning (ML) and deep-learning (DL) are necessary at the beginning of the chapter. There is an embryonic definition between lines 222 and 226 very insufficient. A figure describing in parallel the ML and DL processes will be great.
Could the authors give an example of traditional statistical regression model outperformed by ML? line 194.
In the chapter dedicated to AI in HCC, studies based on non-invasive imagery and those necessitating real histology with biopsies should be separated in different paragraphs and not mixed as it is.
Between lines 227-233, performances of some AI models are given (ROC, accuracy, specificity, sensitivity) but there is no mention of the performances of traditional techniques for comparison. This shortcoming should be corrected.
Concerning microvascular invasion (MVI) the authors indicate from the onset that “there is currently no reliable predictive parameter of MVI). Nevertheless, the author indicate line 244 that radiomics signatures demonstrated superior prognostic probability for HCC recurrence relative to non-radiomics. C_index are provided but not those of previous non-radiomics models. We do not know the magnitude of the progresses.
Concerning treatment, as judged through OS and PFS, it is very difficult to see the “paradigm shift” brought by Durvalumab-Tremelimumab vs Atezolizumab-Bevcizumab (OS 16.4 vs 19.2, PFS 3.8 vs 6.8 respectively). Could the authors explain more clearly where is the “paradigm shift”?
And in general, as “paradigm shift” is the terminology used in title, sentences describing the conceptual nature of these shifts in the three domains (biomarkers, artificial intelligence, and therapy) for the 2 tumor types, HCC and CCA are necessary either at the end of each section or in the general conclusion. Although by reading the manuscript, we clearly see that there are novelties and progresses we are not sure that they represent truly long-lasting shifts in medical practices. Therefore, confronting what will be definitively implemented and what will be definitively abandoned in each domain is necessary to enlighten reader’s mind.
The opinion of the authors about loco-regional therapies (LRT) is unclear. They first present LRT as problematic (line 332) and continue line 339 with RFA as a therapy endowed of high efficacy and low risk complications. It is somewhat difficult to grasp.
A surprising part of the review concerns CCA therapies. It appears clearly that survivals with immunotherapies are much shorter that those with more traditional TACE or TARE approaches. This is looking like more as “paradigm reversal” than a “paradigm shift” but the authors do not insist on this apparent defeat of immunotherapy although it looks conceptually very important. Am I wrong?
Round 2
Reviewer 1 Report
Authors have addressed most of my comments.